# Surface Modification of Silicon Carbide Wafers Using Atmospheric Plasma Etching: Effects of Processing Parameters

**DOI:** 10.3390/mi14071331

**Published:** 2023-06-29

**Authors:** Qi Jin, Julong Yuan, Jianxing Zhou

**Affiliations:** 1Ultra-Precision Machining Centre, Zhejiang University of Technology, Hangzhou 310014, China; 2112002310@zjut.edu.cn (Q.J.); jxzhou@zjut.edu.cn (J.Z.); 2Xinchang Research Institute of ZJUT, Zhejiang University of Technology, Xinchang, Shaoxing 312500, China

**Keywords:** atmospheric plasma, silicon carbide wafe, removal function, volumetric removal rate

## Abstract

Silicon carbide wafer serves as an ideal substrate material for manufacturing semiconductor devices, holding immense potential for the future. However, its ultra-hardness and remarkable chemical inertness pose significant challenges for the surface processing of wafers, and a highly efficient and damage-free method is required to meet the processing requirements. In this study, atmospheric plasma processing was used to conduct point-residence experiments on silicon carbide wafers by varying process parameters such as Ar, CF_4_, and O_2_ flow rate, as well as processing power and the distance between the plasma torch and the workpiece. We investigate the effects of these on the surface processing function of atmospheric plasma etching and technique for surface modification of silicon carbide wafers, evaluating the material removal rates. Then, according to the experimentally derived influence law, suitable parameter ranges were selected, and orthogonal experiments were designed to determine the optimal processing parameters that would enable rapid and uniform removal of the wafer surface. The results indicate that the volume removal rate of the plasma on the silicon carbide wafer achieves its maximum when the input power is 550 W, the processing distance between the plasma torch and workpiece is 3.5 mm, and when the Ar, CF_4_, and O_2_ flow rates are 15 SLM, 70 SCCM, and 20 SCCM, respectively.

## 1. Introduction

As society continues to progress, the fields of optoelectronics and microelectronics are experiencing rapid growth. Semiconductor components are increasingly being used in high-temperature, high-voltage, high-frequency, and high-power operating environments. Silicon carbide (SiC), being a third-generation semiconductor material, possesses superior material properties, including a large forbidden bandwidth, high thermal conductivity, high breakdown electric field, rapid electron saturation rate, excellent thermal stability, and robust radiation resistance [1]. These characteristics make it an ideal substrate material to produce semiconductor devices, and it holds great potential for future advancements in the field. The performance of semiconductor components is significantly influenced by the surface quality of the substrate material [2]. Therefore, it is essential to simplify and optimize the production of silicon carbide wafers to ensure excellent surface quality with minimal damage. However, the unique physicochemical properties of SiC materials, such as their ultra-hardness and remarkable chemical inertness, pose significant challenges for the surface polishing of wafers. These difficulties in achieving efficient and effective polishing processes result in high production costs, which in turn restrict the large-scale application and promotion of SiC materials [3].

After the SiC crystals are grown, they must undergo procedures such as orientation, rounding, and edge grinding to pretreat the SiC ingots prior to slicing, thinning, and polishing to become qualified substrate materials [4,5]. During the slicing and thinning processes, the force applied to the surface of the wafer results in the generation of numerous processing traces and the formation of a substantial layer of subsurface damage [6]. The damage must be eliminated prior to further manufacturing, thus requiring additional polishing [7]. Currently, the most prominent methods used to achieve ultra-smooth silicon carbide wafer surfaces include mechanical polishing, chemical mechanical polishing (CMP) [8], electrochemical mechanical polishing (ECMP) [9,10], and magnetorheological finishing (MRF) [11,12]. However, traditional polishing methods that involve direct contact with the workpiece surface can induce stress and exert forces that inevitably lead to a certain degree of lattice damage on the surface [13,14]. To avoid this problem, plasma machining has emerged as a noncontact machining technique that improves the processing efficiency of removing surface and subsurface damage, resulting in a damage-free workpiece surface [8,15]. In this work, we focus on the development of a polishing method for the plasma processing of silicon carbide wafers to remove surface and subsurface damage while optimizing the material removal rate [16].

## 2. Experimental Principles and Device Details

### 2.1. Plasma Processing Principles

Atmospheric plasma processing is a multiphysical field chemical etching method. Unlike traditional processing methods, atmospheric plasma processing is a noncontact process that can avoid surface and subsurface damage, lattice dislocation, and other defects caused by contact processing.

The reaction process can be divided into three steps. First, the fluorine-based gas is passed into the plasma torch as the reaction gas and excited by high-energy electrons, producing active F ions and fluorine-containing active atomic particles, which are adsorbed on the material surface. Then, the active F particles react chemically with the Si components on the material surface. Finally, the generated volatile gas SiF_4_ overflows from the surface of the workpiece and diffuses rapidly under the action of the jet, leaving a nondamaging surface. Thus, the Si component of the lower material is exposed to contact with the active F particles, and this process is repeated to achieve continuous processing and nondestructive removal of material [17]. The removal process of processing SiC material can be expressed as follows: SiC + F* → SiF_4_↑

### 2.2. Experimental Setup

The experimental device for plasma processing can be divided into three main parts: a plasma generation device, a motion control device, and a gas supply device. These assemblies make up the mechanical structure of the device to achieve the surface processing of workpieces of varying sizes.

The entire machining unit adopted a gantry layout with three linear axes, X, Y, and Z, with strokes of 600, 600, and 400 mm, respectively, with a bidirectional positioning accuracy of 10 μm and repetitive positioning accuracy of 5 μm.

The plasma jet was generated by excitation in the plasma-generating device, and the required working gas was provided by the gas supply device, which was passed into the generating device through a flow controller that precisely controlled the stability of the gas flow. The plasma generator primarily consists of a plasma torch, an RF power supply, and an impedance matcher. The plasma torch is the key component to excite the plasma, which consists of a plasma torch tube, induction coil, nozzle, and electric spark igniter, with the structure shown in Figure 1. After supplying the working gas, a low-frequency power supply with a frequency of 40.68 MHz and an output power range of 5–1000 W is applied to both ends of the coil to generate an induction current, which is ignited by an electric spark igniter to ionize the gas inside the torch tube and produce a stable plasma jet, while the torch tube is cooled by forced air cooling.

The thermal imaging pyrometer is fixed at a certain position from the plasma torch using a fixed bracket, with the lens facing the excitation region of the plasma and perpendicular to the direction of the plasma torch jet, as shown in Figure 2. The pyrometer was used to measure the temperature of the plasma torch and laterally reflect the temperature of the generated plasma jet.

## 3. Experimental Design

The experiments were conducted at room temperature and pressure, using a SiC wafer with a thickness of 350 μm. Ar was selected as the generation gas to excite the plasma, CF_4_ as the reaction gas, and O_2_ as the auxiliary gas. The experiments were conducted by varying process parameters such as the CF_4_, Ar, and O_2_ flow rate, the processing distance between the plasma torch and the workpiece, and the processing power to perform a point dwell on the surface of the SiC wafer.

Before the experiment, the SiC wafer was cleaned in the ultrasonic cleaner for 10 min. It was then taken out and immersed in pure water for soaking and scrubbing. Afterward, it was dried using a dust-free cloth and placed onto the tooling for fixation, and the exhaust gas treatment device was turned on in advance to prepare for the experiment. The experiments were initiated by passing a large amount of Ar into the plasma torch, which was discharged through the RF power supply to ionize Ar to form a stable plasma-active atmosphere as a white flame-like jet. Then, CF_4_ was slowly introduced, and Ar plasma was used to excite the CF_4_ to produce many active F particles, resulting in a green flame. After preheating for 30 min, the plasma torch was controlled to gradually descend close to the surface of the SiC wafer, as shown in Figure 1, allowing the jet to contact and chemically react with the wafer surface, generating the volatile gas SiF_4_ upon material removal. After the dwell processing for a certain time, the operational control system raised the plasma torch away from the wafer surface. The wafer was removed after processing, and roughness was determined via multiple inspections of the machined spot hole profile using a PG1830 roughness profiler (Taylor Hobson, Leicester, UK).

After the plasma jet resides on the surface of the silicon carbide wafer, the removal profile of the processed spot is shown in Figure 3 as a rotationally symmetric Gaussian-type function, which can be expressed by Equation (1):(1)$$r(x,y)=R\cdot e^{-\frac{x^{2}+y^{2}}{2\sigma^{2}}}$$
where R is the removal depth, and σ is a Gaussian distribution parameter.

For ease of detection and expression, the removal function of the rotationally symmetric removal contour is typically described using the bus line, which serves as a representation of the removal contour. A typical removal function contour is shown in Figure 4.

The measured curve was compared with the fitted Gaussian curve using MATLAB software, and the results are shown in Figure 5. A comparison of the two curves shows that the fitting of the Gaussian function has a high degree of agreement.

The plasma removal function is generally described by the peak removal rate and half-height width of the Gaussian function [18,19]. The peak removal rate, *a*, is the magnitude of the removal depth per unit time and is calculated as (2).
(2)$$a=\frac{R}{t}$$

The *FWHM* is the width in the horizontal direction of the removal contour when the removal depth is at $\frac{R}{2}$; it can be expressed as (3).
(3)$$FWHM=2\sqrt{2\ln2}\sigma$$

The material removal rate of SiC by plasma jets is generally described by the volume removal rate *V* of the plasma, which is the volume of material removed by the plasma per unit time and is calculated as (4).
(4)$$V=\iint_{S}ae^{-\frac{x^{2}+y^{2}}{2\sigma^{2}}}dxdy=\int_{0}^{2\pi }\int_{0}^{+\infty }ae^{-\frac{r^{2}}{2\sigma^{2}}}rd\theta dr=2\pi a\sigma^{2}\approx 1.1331aFWHM^{2}$$

The experiments were conducted using the single-factor method by changing only one of the process parameters while keeping the other parameters constant. The effect of each process parameter on the removal function was determined by comparing the peak removal rate *a*, half-height width (*FWHM*), and volume removal rate *V* for different process parameters to explore the removal characteristics of the plasma on the SiC material.

## 4. Results and Discussion

### 4.1. Ar Flow Impact on Removal Function

Owing to the high molecular bonding energy of CF_4_, relatively high power is required for ionization to produce a stable plasma. To continuously excite CF_4_, it is typically passed into a stable plasma environment, and the violent motion of electrons in the excited state of the plasma collides with the CF_4_ molecules to ionize CF_4_ and produce active F particles. Ar was selected owing to its cost-effectiveness and widespread availability compared to other inert gases.

During the processing, although Ar does not directly participate in the plasma chemical reaction, its flow rate plays a crucial role in determining the overall gas flow rate and the characteristics of the plasma jet’s exit, including the flow state and concentration of active particles. Consequently, the Ar flow rate is a significant factor that influences the removal rate in plasma processing. In this experiment, the effect of the Ar flow rate on the removal function was investigated to understand its impact.

The prerequisite for conducting the experiments was maintaining a stable plasma jet state. When the Ar flow rate is too low, the plasma jet becomes weak and may struggle to efficiently excite the plasma. Conversely, when the Ar flow rate is too high, it can disrupt the plasma jet and compromise the uniformity of the discharge. The plasma torch can produce a stable plasma jet in the range of an Ar flow rate of 15–27 SLM. Under the experimental parameters shown in Table 1, the experiment was divided into six levels with Ar flow rates of 15, 17, 19, 21, 23, 25, and 27 SLM, with the other parameters fixed, and a point dwell was performed on the wafer surface to investigate the variation law of the removal function.

The experimental results are presented in Figure 6. The Ar flow rate has a small effect on the *FWHM* of the removal function, and the peak removal rate *a* decreased with increasing Ar flow rate, whereas the volume removal rate *V* was affected by both the *FWHM* of the removal function and the peak removal rate *a*, which also decreased gradually with increasing Ar flow rate. The observed decrease in the peak removal rate and volume removal rate with increasing Ar flow rate can be attributed to the need to maintain a stable plasma jet state. To prevent the plasma jet from ceasing owing to the “avalanche effect” after the introduction of CF_4_ and O_2_, a sufficient or excessive incoming Ar flux is often utilized to ensure the excitation of enough gas. In the experiments, when the Ar flow rate was relatively low, the green color of the plasma jet was darker, indicating a higher concentration of active F particles in the jet. Therefore, the peak removal rate *a* and volume removal rate *V* of the removal function were also larger. However, as the Ar flow rate continued to increase, the jet color gradually became lighter, suggesting an increasing excess of Ar and a subsequent reduction in the concentration of CF_4_-excited active F particles. A lower concentration of active F particles leads to a decrease in the peak removal rate *a* and, consequently, a decrease in the volume removal rate *V*.

### 4.2. CF_4_ Flow Impact on Removal Function

The essence of plasma etching of SiC wafers is to remove the wafer surface material using the chemical reaction between the reactive F particles in the plasma jet and the silicon carbide surface. CF_4_ excitation to generate reactive F particles in the reaction gas is the core of plasma processing. The CF_4_ flow rate is the key factor in determining the concentration of reactive F particles within the plasma jet. The effect of different CF_4_ flow rates on the removal function was examined by varying the CF_4_ flow rates while keeping other parameters constant. The CF4 flow rates investigated in the experiments were 0, 15, 25, 35, 45, 55, 65, 75, and 85 SCCM, as shown in Table 2. The experimental results are illustrated in Figure 7.

As can be seen from Figure 7, the plasma jet does not etch the SiC wafer when the reaction gas CF_4_ is not passed inside the excited gas, indicating that active F particles are necessary for the plasma etching provided by CF_4_.

When the CF_4_ flow rate was <65 SCCM, the *FWHM* of the removal function did not change significantly and fluctuated at a fixed value, whereas the peak removal rate *a* and peak removal rate *V* increased with the flow rate. This is because the CF_4_ flow rate was very low, nearly one-thousandth of the Ar flow rate. Although the CF_4_ flow rate gradually increased, the total flow rate of the gas excited to produce the plasma was basically constant, resulting in a constant flow field and *FWHM.* As the flow rate of the reaction gas gradually increased, the concentration of active F particles increased. This, in turn, promoted the chemical reaction rate between the plasma and the surface of the SiC. Consequently, the peak removal rate *a* of the removal function increase and obtains a higher volume removal rate *V*.

However, when the CF_4_ flow rate continues to increase, the *FWHM* and volume removal rate *V* of the removal function gradually decreases, and the rate of increase in the peak removal rate *a* decreases. CF_4_ poses challenges in direct excitation by the electric field to form plasma. Furthermore, CF_4_ itself does not undergo chemical reactions with SiC material. Consequently, high CF_4_ flow rates can affect the plasma atmosphere generated by Ar and narrow the plasma discharge range, thereby limiting the excitation of CF_4_. Therefore, the *FWHM* decreases in the removal function, causing the volume removal rate *V* to decrease.

Therefore, the volume removal rate *V* of the plasma jet was maximum at an input power of 500 W, Ar flow rate of 19 SLM, and CF_4_ flow rate of approximately 65 SCCM.

### 4.3. O_2_ Flow Impact on Removal Function

O_2_ is not directly involved in the chemical reaction of the etched silicon carbide material during processing; however, the passage of an appropriate amount of O_2_ can enhance the processing rate of plasma-etched SiC while simultaneously reducing the deposit generation. Single-factor experiments were conducted using the test parameters listed in Table 3 to study the effect of O_2_ flow rate on the removal function, with experimental results shown in Figure 8.

As shown in Figure 8, when the plasma was excited by auxiliary gas O_2_ below 20 SCCM, the peak removal rate, *a,* and volume removal rate, *V*, of the process removal function were lower than those of the plasma excited by Ar and CF_4_ only. This indicates that the inhibitory effect of O_2_ on the active F particle production was greater than the generation effect when the incoming O_2_ flow rate was lower than 20 SCCM. The peak removal rate, *a*, and *FWHM* of the removal function increased significantly, and the volume removal rate, *V*, increased when the O_2_ flux was gradually increased. The volume removal rate, *V*, was the largest at a rate of 40 SCCM O_2_, which was more than twice the volume removal rate, *V*, without O_2_.

It can be found that the removal function and the volume removal rate, *V*, are similar between O_2_ flow rates of 20 and 0 SCCM. The plasma processing spot removal profile bus diagram on the surface of the SiC wafer under the two process parameters was observed separately, as shown in Figure 9. The maximum processing diameters of the two processing spots are close to each other at approximately 4 mm. The two processing spots were examined separately using a super-depth-of-field microscope, and the results are shown in Figure 10.

Among them, Figure 10a presents the detection map of the plasma processing spot without the introduction of O_2_ auxiliary gas. The presence of a distinct circular layer of white deposition surrounding the processing spot around the center is clearly evident in the figure. The visible deposition range is approximately 7.5 mm in diameter, which is much larger than the actual processing spot removal diameter. Furthermore, the white deposition deteriorates the surface roughness of the wafer, which requires removal by a subsequent fine polishing process. Figure 10b shows the processing spot detection when 20 SCCM of O_2_ was applied. The deposition is significantly reduced, and the overall deposition circle with the processing center as the origin is approximately 6.8 mm in diameter. Moreover, the visible deposition range is approximately 0.7 mm smaller than the deposition range formed when the auxiliary gas O_2_ is not introduced. The amount of deposition was significantly reduced inside the ring. A comparison between the two graphs reveals that the introduction of O_2_ in the working gas has a significant impact. It reduces the extent of deposition and minimizes the formation of large white deposits caused by the etching reaction. This leads to an improvement in the surface quality of the processed workpiece and a reduction in the time required for fine polishing. Overall, the inclusion of O_2_ enhances the processing efficiency of SiC wafers. 

This can be attributed to the reaction of O with particles such as CF_2_, CF_3_, and COF generated by CF_4_ within the plasma. This reaction gives rise to compounds such as COF_2_ and OF*. Notably, OF* readily decomposes to produce F*, increasing the number of active F particles and promoting the chemical reaction rate. However, the deposits were mainly due to the large number of sedimentary particles generated after CF_4_ is ionized, such as CF, CF_2_, CF_3_, etc. These fluorocarbons reach the workpiece surface with the plasma jet and combine with the Si ions in the material to form polymeric macromolecules attached to the workpiece surface, which block further etching by active F particles. The incoming O reacts with all three particles, which inhibits the generation of deposits and improves the etching efficiency. However, because O must consume the F atoms in the plasma when reacting with the generated COF_2_ particles, the etching reaction was inhibited to some extent. Therefore, it is necessary to control the O_2_ flow rate and experimentally determine a suitable O_2_ flow rate range.

Observation of the plasma jet morphology during processing showed that when the incoming O_2_ flow rate increased to 20 SCCM, the plasma jet shape started to become thinner and narrower, and the visible length of the jet decreased. Additionally, the color gradually became lighter, from green to white, as shown in Figure 11. According to Figure 8b, the *a* of the removal function continues to increase rapidly, indicating that the effective processing distance of the plasma jet far exceeds the visible jet length; that is, the effective action distance of the active particles is much larger than the visible jet length.

When the O_2_ flow rate exceeds 40 SCCM, the peak removal rate of the removal function rises at a slower rate. The *FWHM* fluctuates at a fixed value, and the volume removal rate remains relatively constant. However, it is observed that the plasma jet starts to flicker or even extinguish. This indicates that an increase in the O_2_ flow rate causes plasma instability. Therefore, careful control of the O_2_ flow rate is necessary during processing to maintain stable plasma conditions.

### 4.4. Effect of Processing Power on Removal Function

Processing power is the magnitude of the power supplied to both ends of the induction coil by the RF power supply during plasma processing. It serves as the energy source for the plasma system and plays a crucial role in determining the effectiveness of the processing. The maintenance of plasma is dependent on the continuous oscillation collision of electrons in the excited state with gas molecules so that the compound ionization process occurs continuously in the plasma. However, because electrons gradually lose energy during the collision process with gas molecules, the RF power supply is needed to continuously excite new electrons to continue the reaction.

The effect of machining power on the removal function was experimentally studied by varying machining power in the 350–650 W range with a step size of 50 W and other experimental parameters, as listed in Table 4.

The experimental results shown in Figure 12 indicate that the *FWHM*, peak removal rate, and volume removal rate of the removal function all increase linearly with an increase in processing power, and the changes are significant. In the case of an adequate supply of processing gas, an increase in machining power results in higher energy input to the plasma. This leads to an increase in electron density within the plasma, which in turn excites more CF_4_ gas and raises the concentration of active F particles. As a result, the peak removal rate (*a*) of the removal function increases. Additionally, the increase in electron energy expands the diffusion range of active F particles, leading to an increase in the *FWHM* of the removal function. Thus, the relationship between machining power and the removal function is characterized by an increase in both the peak removal rate (a) and the *FWHM* as machining power increases.

The processing power input to an RF power supply cannot be increased indefinitely. During the experiments, when a power of 700 W was used to excite the plasma for processing the SiC wafer, it resulted in the wafer breaking. It was observed that variations in the processing parameters led to differences in the temperature of the plasma generated by excitation. The plasma temperature is an important factor that affects the etching rate of the plasma. The available literature and preliminary experimental studies show that processing power plays a decisive role in determining the magnitude of the plasma temperature [20]. Studies show that the plasma temperature increases linearly with power, and the higher the processing power, the faster the plasma temperature increases. A higher processing power can increase the temperature of the chemical reaction between the active F particles and silicon carbide, promoting the rate of the etching reaction. Consequently, this reduces the processing time for removing surface and subsurface damage. Conversely, a smaller processing power can reduce the temperature of the plasma and reduce the deformation of the wafer material due to heat. Therefore, in actual processing, the relationship between the material removal rate and plasma temperature should be balanced and the processing power optimized.

### 4.5. Effect of Processing Distance on Removal Function

From a macroscopic perspective, a plasma jet can be likened to a fluid. As it emerges from the nozzle, the jet initiates a divergence, becoming more pronounced as it moves away from the nozzle. Generally, the distance from the plasma torch nozzle to the workpiece surface is referred to as the processing distance. The experiments were conducted at a processing power of 500 W, and the processing distances were varied and set to 3, 4, 5, 6, and 7 mm, with other experimental parameters held constant, as shown in Table 5.

The experimental results are shown in Figure 13. The larger the processing distance between the nozzle and the SiC wafer, the smaller the peak removal rate of the removal function. It was observed that this relationship is not strictly linear. When the machining distance increased from 3 to 5 mm, the *FWHM* of the removal function did not change significantly, and the volume removal rate decreased gradually. However, when the machining distance increased to 6 mm, the *FWHM* and volume removal rate decreased sharply; when the machining distance increased to 7 mm, the *FWHM* and volume removal rate did not change substantially. This indicates that as the processing distance increased, the degree of jet dispersion increased, resulting in a relative decrease in the number of active F particles diffused to the surface by the jet. In the experimental setup conducted in an atmospheric environment, it was observed that once the plasma jet exits the nozzle and interacts with the surrounding air, the active particles within the plasma collide with the neutral gas molecules in the air, resulting in energy loss. This energy loss becomes more significant as the processing distance increases, leading to a decrease in the chemical etching capability of the plasma.

Comparing the measured contour curves of the removal function, it can be found that at a low processing distance, the measured contour curves are very smooth and fit the Gaussian curve, as shown in Figure 14. When the processing distance is further, some small fluctuations in the contour curves deviate slightly from the Gaussian curve, as shown in Figure 15. This phenomenon could be attributed to the susceptibility of the plasma jet to the surrounding airflow as the machining distance increases. The influence of the airflow can result in instability in the shape of the plasma jet and cause slight variations in the contour curve.

### 4.6. Experimental Exploration of Optimal Processing Parameters

To improve the processing rate of SiC wafers via atmospheric plasma polishing, it is necessary to determine the optimal processing parameters. Due to the multitude of process parameters that influence plasma processing efficiency and the intricate interrelationships among these factors, conducting preliminary single-factor experiments only examines the impact of a single process parameter. Consequently, this approach entails certain limitations. It fails to account for the complex coupling effects that may arise from the interactions between multiple process parameters, leading to an incomplete understanding of the overall system behavior. To study the influence of multiple factors on processing and to investigate the optimal processing parameters for processing SiC wafers, orthogonal experiments were designed and conducted based on previous experimental findings.

The factors that affect the plasma removal function and volume removal rate are the Ar, CF_4_, and O_2_ flow rates, processing power, and processing distance. Four levels of these five factors were taken separately for experimental analysis, and the L_16_(4^5^) orthogonal experimental table was selected for the design. A total of 16 sets of experiments were conducted, as shown in Table 6. The same PG1830 roughness profiler was used to detect the removal profile of the machined hole, calculate the *FWHM* and peak removal rate, and derive the volume removal rate of the plasma for this parameter.

The volume removal rate is a more intuitive representation of the magnitude of the material removal rate of the plasma from the SiC wafer and is the most critical index for evaluating plasma processing. Therefore, orthogonal experiments were mainly performed on the volume removal rate results for the extreme difference analysis, and K_1_, K_2_, K_3_, K_4_, k_1_, k_2_, k_3_, k_4_ and the extreme value R corresponding to each factor were calculated, collated, and analyzed, as shown in Table 7.

Comparing the five extreme differences in R under the volume removal rate index, it can be found that the main order of influence of each process parameter on the volume removal rate of silicon carbide processed by plasma was processing power (4.117), Ar flow rate (4.05), O_2_ flow rate (3.719), processing distance (3.351), and CF_4_ flow rate (2.337). The optimal process parameters to improve the machining efficiency were a power of 550 W, a machining distance of 3.5 mm, an Ar flow rate of 15 SLM, a CF_4_ flow rate of 70 SCCM, and an O_2_ flow rate of 20 SCCM, with the highest volume removal rate for this process parameter. This combination did not occur in the 16 machining experiments, which also reflects the advantages of orthogonal experiments.

The plasma jet excited with this processing parameter processed the silicon carbide wafer with the processing spot profile curve shown in Figure 16. The measurement shows that the peak removal rate of the plasma removal function at this parameter is 0.5736 μm, and the half-height width is 3.456 μm. The volume removal rate is calculated to be 9.119 μm^3^/s. The comparison shows that the volume removal rate of the removal function at this parameter is the largest and is the optimal processing parameter.

## 5. Conclusions

Single-factor experiments were conducted to investigate the impact of Ar flow rate, CF_4_ flow rate, O_2_ flow rate, processing power, and processing distance on the removal function and volume removal rate during plasma processing of SiC wafers. The purpose of these experiments was to gain a deeper understanding of the underlying reasons behind these effects:(1)The Ar flow rate had a small effect on the half-height width of the removal function, and the peak and volume removal rates decreased with an increasing Ar flow rate. Therefore, to improve the volumetric removal rate of the material during processing, the incoming Ar flux should be minimized while maintaining a stable plasma je.(2)Active F particles are necessary for the plasma etching of silicon carbide wafers, and CF_4_ is the only source of active F particles. The volume removal rate of the material increased with the CF_4_ flow rate and then decreased. The volume removal rate of the plasma jet was maximum at an input power of 500 W and an Ar flow rate of 19 SLM, and the CF_4_ flow rate was approximately 65 SCCM.(3)The introduction of the auxiliary gas O_2_ destabilizes the plasma jet, but an appropriate amount of O_2_ can improve the plasma processing efficiency. When the input power was 500 W, the Ar flow rate was 19 SLM, the CF_4_ flow rate was 60 SCCM, and the incoming 20–40 SCCM of O_2_ increased the etching reaction efficiency.(4)The half-height width, peak removal rate, and volume removal rate of the removal function all increased linearly with an increase in processing power and changed significantly. However, excessive processing power increases the plasma temperature, leading to a rapid increase in the temperature of the silicon carbide in contact with the jet region, and the wafer undergoes deformation by heat or even shattering.(5)The degree of plasma dispersion increases with the processing distance, such that the diffusion of active F particles to the silicon carbide surface is relatively reduced; thus, the processing distance between the plasma nozzle and the silicon carbide wafer should be less than 5 mm.(6)Based on the results of the previous experiments, orthogonal experiments were designed to determine the optimal process parameters for the plasma processing of SiC wafers. It was found that the maximum volumetric removal rate of silicon carbide by plasma was achieved when the power input power was 550 W, the processing distance between the plasma torch and the workpiece was 3.5 mm, the incoming Ar flow rate was 15 SLM, CF_4_ flow rate was 70 SCCM, and the O_2_ flow rate was 20 SCCM.

## Figures and Tables

**Figure 1 micromachines-14-01331-f001:**
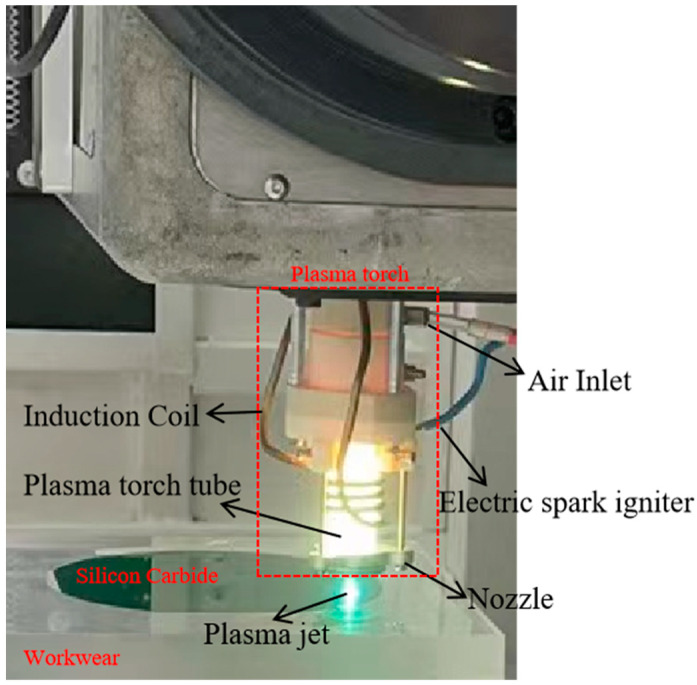
Plasma torch processing diagram.

**Figure 2 micromachines-14-01331-f002:**
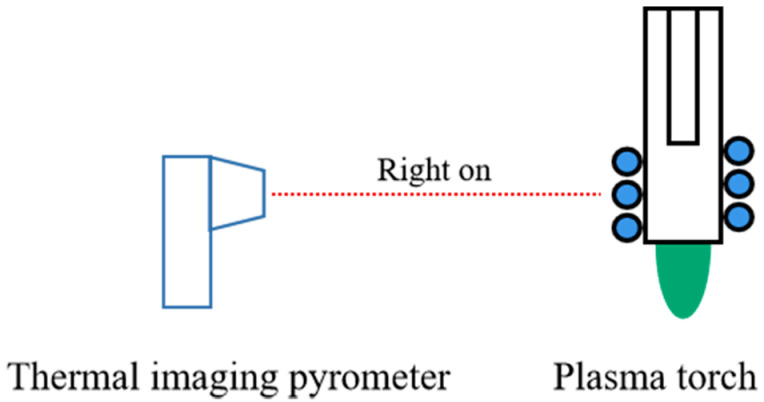
Thermal imaging pyrometer measurement schematic.

**Figure 3 micromachines-14-01331-f003:**
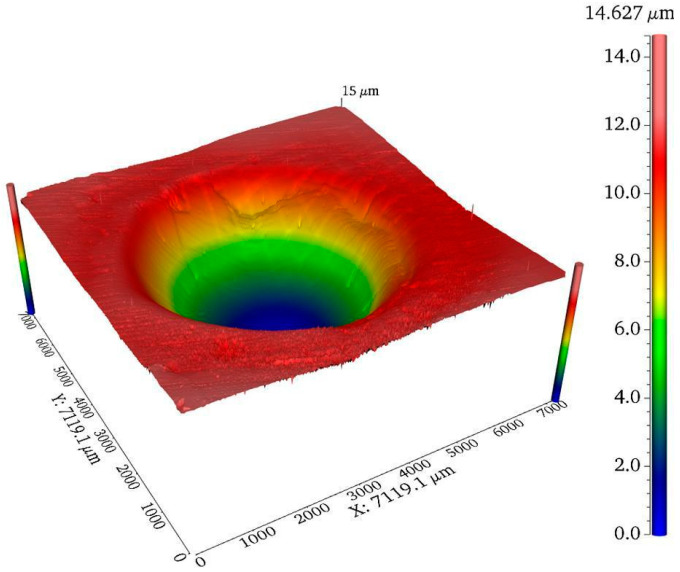
Processing spot removal contour.

**Figure 4 micromachines-14-01331-f004:**
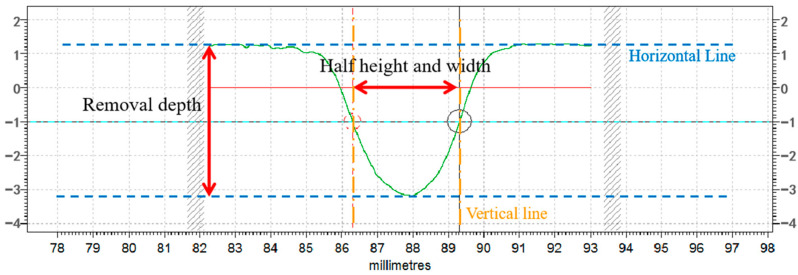
Removal of function profile bus.

**Figure 5 micromachines-14-01331-f005:**
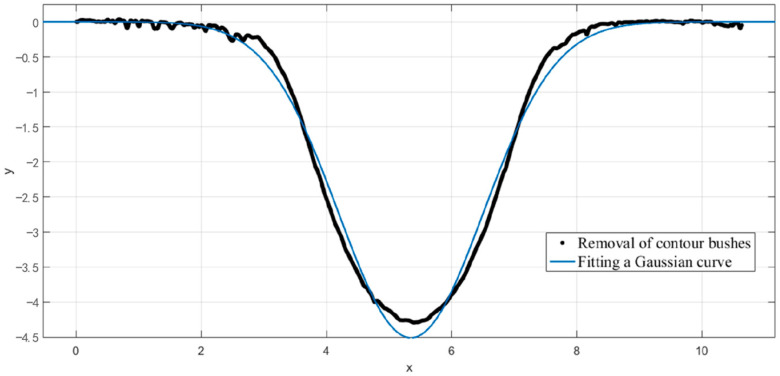
Removing contours vs. fitting curves.

**Figure 6 micromachines-14-01331-f006:**
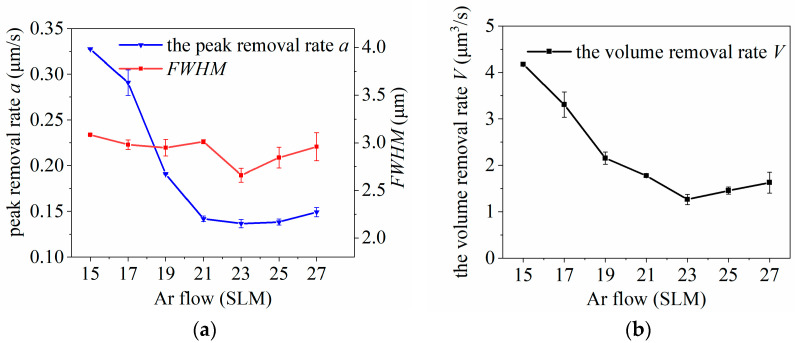
Effect of Ar flow variation on plasma processing: (**a**) influence curve of Ar flow variation on removal function; (**b**) influence curve of Ar flow rate change on volume removal rate.

**Figure 7 micromachines-14-01331-f007:**
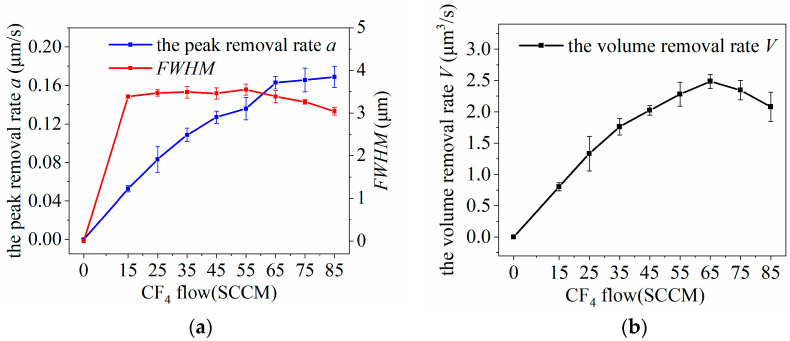
Effect of CF_4_ flow variation on plasma processing: (**a**) influence curve of CF_4_ flow variation on removal function; (**b**) influence curve of CF_4_ flow rate change on volume removal rate.

**Figure 8 micromachines-14-01331-f008:**
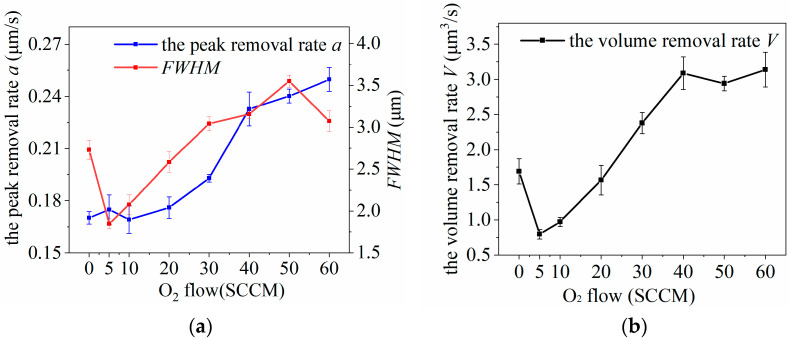
Effect of O_2_ flow variation on plasma processing: (**a**) influence curve of O_2_ flow variation on removal function; (**b**) influence curve of O_2_ flow rate change on volume removal rate.

**Figure 9 micromachines-14-01331-f009:**
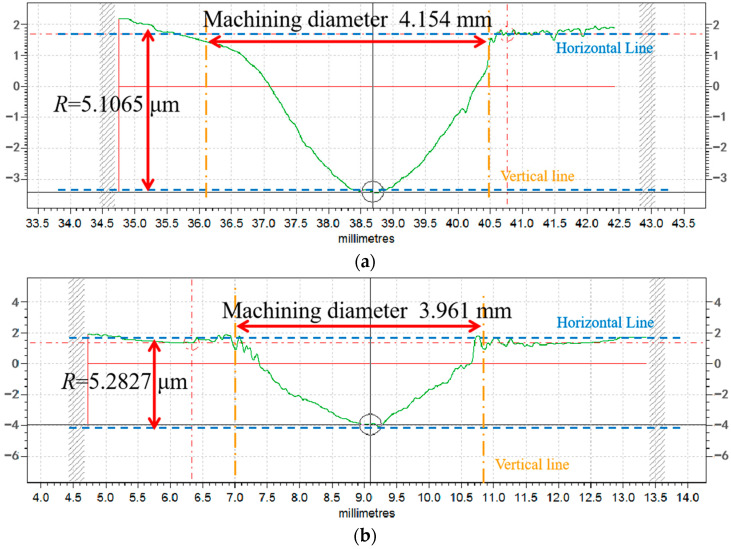
Machining spot profile bus diagram at different O_2_ flow rates: (**a**) removal of contour bus diagram when O_2_ flow rate is 0 SCCM; (**b**) removal of contour bus diagram when O_2_ flow rate is 20 SCCM.

**Figure 10 micromachines-14-01331-f010:**
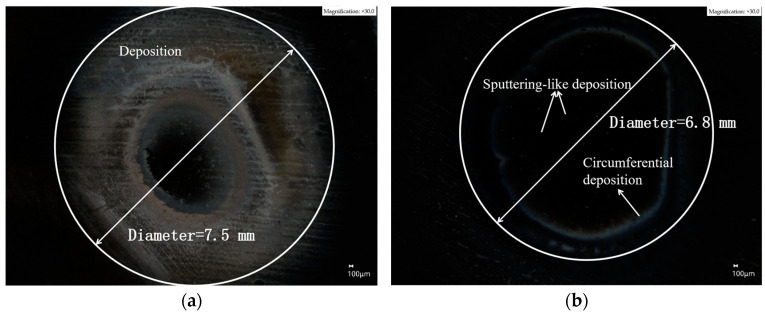
Comparison of surface deposition of processed spots at different O_2_ flow rates: (**a**) O_2_ flow rate of 0 SCCM at the ultra-deep field microscope detection map; (**b**) O_2_ flow rate of 20 SCCM at the ultra-deep field microscope detection map.

**Figure 11 micromachines-14-01331-f011:**
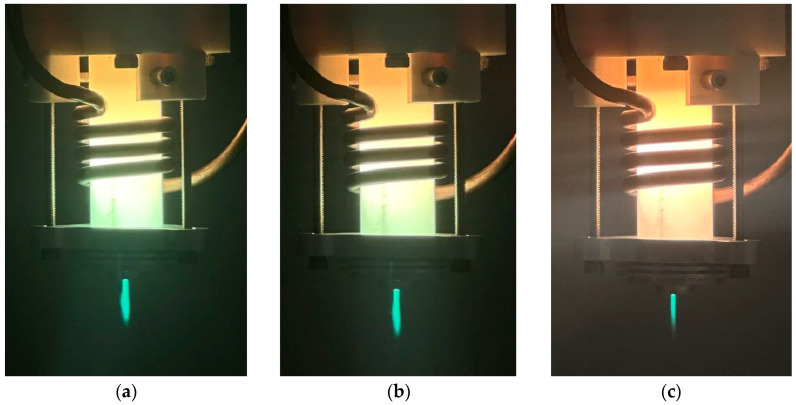
Plasma jet morphology at different O_2_ flow rates: (**a**) 0 SCCM; (**b**) 10 SCCM; (**c**) 20 SCCM; (**d**) 30 SCCM; (**e**) 40 SCCM.

**Figure 12 micromachines-14-01331-f012:**
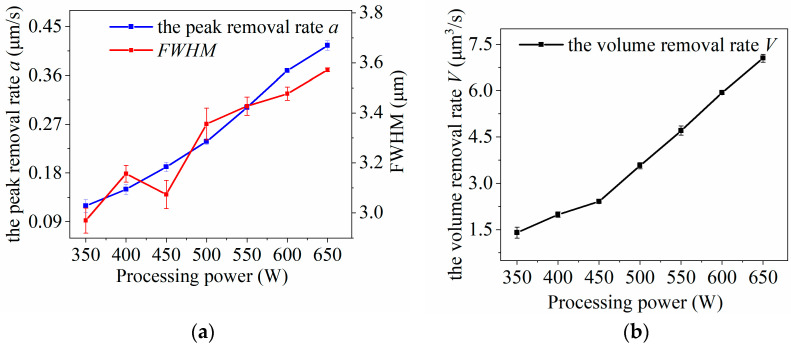
Effect of processing power on plasma processing: (**a**) influence curve of machining power variation on removal function; (**b**) influence curve of processing power variation on volume removal rate.

**Figure 13 micromachines-14-01331-f013:**
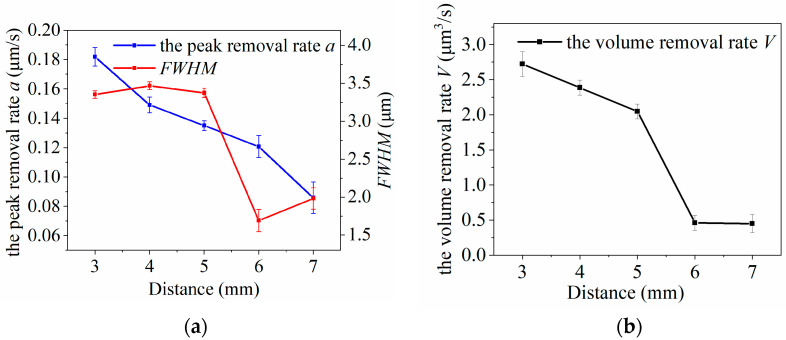
Effect of processing distance on plasma processing: (**a**) influence curve of machining distance change on removal function; (**b**) influence curve of processing distance variation on volume removal rate.

**Figure 14 micromachines-14-01331-f014:**
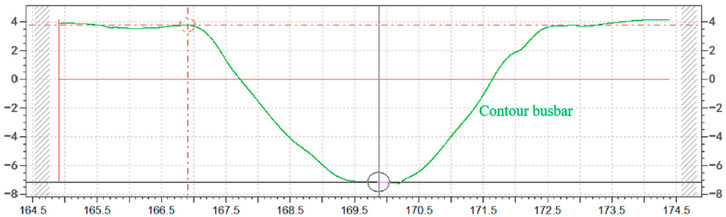
Three-millimeter machining distance for removal of contours.

**Figure 15 micromachines-14-01331-f015:**
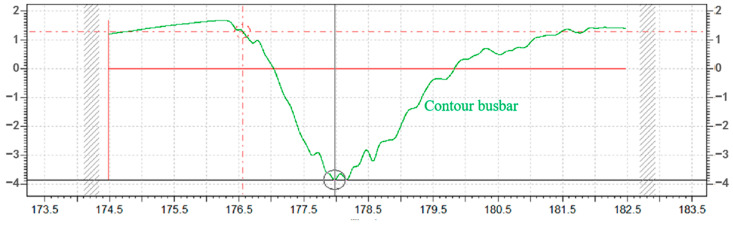
Seven-millimeter machining distance for removal of contours.

**Figure 16 micromachines-14-01331-f016:**
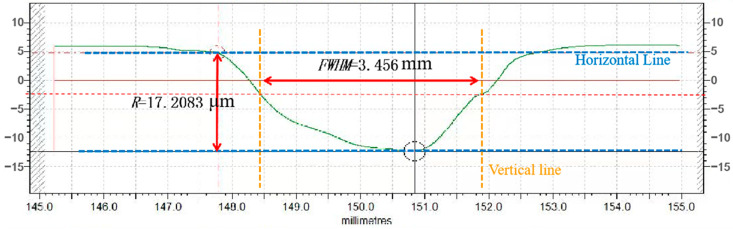
Contour curve of removal function with optimal parameters.

**Table 1 micromachines-14-01331-t001:** Ar flow experimental parameters.

Ar/SLM	Processing Power/W	Time/s	Processing Distance/mm	CF_4_/SCCM
15, 17, 19, 21, 23, 27	500	30	4	60

**Table 2 micromachines-14-01331-t002:** CF_4_ flow experimental parameters.

Ar/SLM	Processing Power/W	Time/s	Processing Distance/mm	CF_4_/SCCM
19	500	30	4	0, 15, 25, 35, 45, 55, 65, 75, 85

**Table 3 micromachines-14-01331-t003:** O_2_ flow experimental parameters.

Ar/ SLM	Processing Power/W	Time/s	Processing Distance/mm	CF_4_/ SCCM	O_2_/SCCM
19	500	30	4	60	0, 5, 10, 20, 30, 40, 50, 60

**Table 4 micromachines-14-01331-t004:** Processing power experimental parameters.

Ar/SLM	Processing Power/W	Time/s	Processing Distance/mm	CF_4_/SCCM
19	350, 400, 450, 500, 550, 600, 650	30	4	60

**Table 5 micromachines-14-01331-t005:** Processing distance experimental parameters.

Ar/SLM	Processing Power/W	Time/s	Processing Distance/mm	CF_4_/SCCM
19	500	30	3, 4, 5, 6, 7	60

**Table 6 micromachines-14-01331-t006:** Orthogonal test design table.

Number	Processing Power/W	Processing Distance/mm	Ar/ SLM	O_2_/SCCM	CF_4_/SCCM	FWHM/μm	*a*/μm	*V/*μm^3^/s
1	450	3.5	15	30	40	3.68	0.476	8.577
2	450	4	17	20	50	4.526	0.221	6.026
3	450	4.5	19	10	60	2.171	0.2	1.256
4	450	5	21	0	70	1.932	0.14	0.697
5	500	3.5	17	10	70	2.585	0.347	3.086
6	500	4	15	0	60	2.951	0.275	3.186
7	500	4.5	21	30	50	3.072	0.129	1.614
8	500	5	19	20	40	2.734	0.144	1.435
9	550	3.5	19	0	50	3.783	0.307	5.857
10	550	4	21	10	40	3.072	0.246	3.095
11	550	4.5	15	20	70	3.619	0.708	12.349
12	550	5	17	30	60	3.142	0.341	4.485
13	600	3.5	21	20	60	3.322	0.462	6.79
14	600	4	19	30	70	3.444	0.566	8.932
15	600	4.5	17	0	40	3.049	0.239	2.952
16	600	5	15	10	50	2.857	0.395	4.287

**Table 7 micromachines-14-01331-t007:** Volumetric removal rate extreme difference analysis.

	Processing Power/W	Processing Distance/mm	Ar/SLM	O_2_/SCCM	CF_4_/SCCM
K1	16.556	24.31	28.399	12.692	16.059
K2	9.32	21.238	16.549	11.725	17.784
K3	25.786	18.172	17.479	26.6	15.717
K4	22.962	10.904	12.197	23.608	25.064
k1	4.139	6.007	7.1	3.173	4.015
k2	2.33	5.31	4.137	2.931	4.446
k3	6.447	4.543	4.137	6.65	3.929
k4	5.741	2.726	3.049	5.902	6.266
R	4.117	3.351	4.05	3.719	2.337
Order of Influence	processing power > Ar > O_2_ > processing distance > CF_4_				
Optimal level	550	3.5	15	20	70

## Data Availability

Data are only available upon request due to restrictions regarding, e.g., privacy and ethics. The data presented in this study are available from the corresponding author upon request. The data are not publicly available due to their relation to other ongoing research.

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
