# Peer review of "Surface Modification of Silicon Carbide Wafers Using Atmospheric Plasma Etching: Effects of Processing Parameters"

_micromachines, 2023, doi:10.3390/mi14071331_

Round 1

Reviewer 1 Report

This work provides some technological insights onto surface modification of silicon carbide wafers through plasma processing. I have some comments as shown below:

1. A more attractive and relevant title can help to further highlight the work. For example “Surface modification of silicon carbide wafers using atmospheric plasma etching: Effects of processing parameters”

2. English language of the article can be improved. Below there are just few examples

 For example, it the introduction section, it is mentioned that “.. processing of wafers, and a highly efficient and damage-free processing method is needed to meet the processing requirements.” This statement can be edited to read “… processing of wafers. Therefore, more efficient and less damaging methods are needed to meet the processing requirements”.

3. Abstract, line 12: “In the study” should read “In this study”

4. Abstract: Please revise lines 14-17. For example: “In this study, atmospheric plasma processing was used to conduct point-residence experiments on silicon carbide wafers by varying process parameters such as Ar, CF4, and O2 flow rates, as well as processing power and the distance between the plasma torch and the work piece. We investigate the effects of these on the surface processing function of atmospheric plasma etching technique for surface modification of silicon carbide wafers, evaluating the material removal rates.

5. Please provide error bars for graphs (such as those shown in figures 7-9 and 13) where possible.

6. The results should be further compared with the literature, and their implications discussed. As the results, I would suggest to expand the reference list, as currently only 9 references are cited in the article.

Moderate English edit of the text is required.

Reviewer 2 Report

This manuscript investigated the influence of atmospheric plasma etching of silicon carbide by varying different process parameters. Single factor experiments of the process parameters were used to analyze the influence of each parameter on the removal function and volume removal rate, but the manuscript needs further improvement. This reviewer has some important comments as follows:

1.     The introduction section of the paper does not provide a comprehensive introduction to SiC processing methods and further literature research should be undertaken. Below are references on etching (International Journal of Extreme Manufacturing, 2021, 3(3): 35104, Small Methods, 2022, 6:2200329) may be of some help. Authors are encouraged to include them in the introduction.

2.     A parameter description is missing in line 130 of the manuscript.

3.     In this paper, a is the peak removal rate and V is the volume removal rate. Provide a reference or explain the reason for using these parameters.

4.     Figure 11 is blurred. It is difficult to get useful information from such an image. Please replace it with one of higher resolution.

5.     All graphs use peak removal rate a and volume removal rate V, but only abbreviations appear in all graphs. These abbreviations are very confusing and it is recommended to use full names and abbreviations in the graph.

6.     This paper studies the effect of a single processing parameter variation on the morphology of plasma etched SiC by the single factor experimental method, but the relationship between the different processing parameters is not obvious. Therefore, this manuscript is more of an experimental report than a scientific research paper.

7.     The author studied the effects of various parameters on the plasma etching of SiC and obtained an optimal parameter-process combination, but the results of using this parameter to process SiC are not reflected in the manuscript.

not applicable

Reviewer 3 Report

In this manuscript, Jin et al reported the silicon carbide wafers atmospheric plasma etching, the paper can be accepted after the following issue were concerned.

1. The surface roughness of the silicon carbide should be provided through the AFM test.

2. The error bar should be added in some of the test.

3. The mechanism section should be enhanced.

4. It is noticed O2 flow rate plays an important part here, what is the surface function changed, can the authors further provided XPS test?

meet requirement.

Reviewer 4 Report

Dear Authors

I was reading your work and the paper with doi=https://doi.org/10.3390/mi14050992, where temperature accumulation is being thoroughly explained.

I think that your work needs to include the above mentioned work and make it clear to the readers the finding of your works in terms  of all parametes, temperature, gas/type flow etc

An example from the abstract: "The results show that the volume removal rate of the removal function can be improved and higher material removal rate can be obtained by passing 15 slm of Ar, about 65 sccm of CF4, 20-40 sccm of O2, processing distance less than 5 mm and increasing the input power of the RF power supply as much as possible without breaking the wafer."

The "processing distance" does not make sense as English wording

Author Response

请参阅附件

Round 2

Reviewer 2 Report

It is clear that the authors did not read the reviewers' comments, e.g. comment1 and comment7, and that the layout and formatting of the article needs further improvement to enhance readability.

Minor editing of English language required

Reviewer 4 Report

Dear Authors

I believe that the manuscript is now ready for acceptance. Thank you for the nice work,

English is fine!